# Microbial Precipitation of Pb(II) with Wild Strains of *Paraclostridium bifermentans* and *Klebsiella pneumoniae* Isolated from an Industrially Obtained Microbial Consortium

**DOI:** 10.3390/ijms232012255

**Published:** 2022-10-14

**Authors:** Olga Neveling, Thato M. C. Ncube, Ziyanda P. Ngxongo, Evans M. N. Chirwa, Hendrik G. Brink

**Affiliations:** Water Utilisation and Environmental Engineering Division, Department of Chemical Engineering, University of Pretoria, Lynnwood Road, Hatfield, Pretoria 0002, South Africa

**Keywords:** lead, bioprecipitation, *K. pneumoniae*, *P. bifermentans*, bioremediation, wastewater

## Abstract

The study focused on determining the microbial precipitation abilities of bacterial strains that were isolated from an industrially obtained Pb(II)-resistant microbial consortium. Previous research has demonstrated the effectiveness of the consortium on the bioprecipitation and adsorption of Pb(II) from solution. The bioremediation of Pb(II) using microbial precipitation provides an alternative option for Pb(II) removal from wastewater. Both strains, *Klebsiella pneumoniae* and *Paraclostridium bifermentans*, were successfully isolated from the consortium obtained from a battery recycling plant in South Africa. The experiments were conducted over both 30 h and 5 d, providing insight into the short- and long-term precipitation abilities of the bacteria. Various initial concentrations of Pb(II) were investigated, and it was found that *P. bifermentans* was able to remove 83.8% of Pb(II) from solution with an initial Pb(II) concentration of 80 mg L${}^{-1}$, while *K. pneumoniae* was able to remove 100% of Pb(II) with the same initial Pb(II) concentration after approximately 5 d. With the same initial Pb(II) concentration, *P. bifermentans* was able to remove 86.1% of Pb(II) from solution, and *K. pneumoniae* was able to remove 91.1% of Pb(II) from solution after 30 h. The identities of the precipitates obtained for each strain vary, with PbS and Pb${}^{0}$ being the main species precipitated by *P. bifermentans* and PbO with either PbCl or Pb${}_{3}$(PO${}_{4}$)${}_{2}$ precipitated by *K. pneumoniae*. Various factors were investigated in each experiment, such as metabolic activity, nitrate concentration, residual Pb(II) concentration, extracellular and intracellular Pb(II) concentration and the precipitate identity. These factors provide a greater understanding of the mechanisms utilised by the bacteria in the bioprecipitation and adsorption of Pb(II). These results can be used as a step towards applying the process on an industrial scale.

## 1. Introduction

The occurrence of lead pollution in wastewater is an area of concern due to the health risks associated with the consumption of lead. These risks include neurodevelopmental alterations, disruption of cellular metabolism through bonding of essential molecules, including haemoglobin synthesis and neurodegeneration, and the occurrence of diarrhoea, stomach pain, kidney damage and anaemia [1,2,3]. Lead is introduced to wastewater through anthropogenic sources such as fertilisers, battery waste, pesticides and industrial wastewater effluents [4]. It is estimated that environmental Pb(II) concentrations have increased by three orders of magnitude in the last 300 years due to these anthropogenic activities [5]. The presence of lead in wastewater is significant in South Africa where the amount of lead in sewage sludge of various municipalities in Limpopo province has been recorded to be higher than allowable amounts. These amounts reached values of 102.8 mg dissolved Pb per kg dry mass in the city of Polokwane and 171.9 mg dissolved Pb per kg dry mass in the town of Louis Trichardt [6]. These values are higher than the allowable limit of 100 mg dissolved Pb per kg dry mass as reported by the Water Resources Commission, South Africa [7]. The conventional methods of lead removal include ion exchange, electrowinning, electrocoagulation, cementation, reverse osmosis, electrodialysis, chemical coagulation and precipitation [8,9]. These methods have the tendency to be expensive and limited in their uses; these limitations include the generation of sludge, a low retention of metal ions, a low selectivity for metals, and high energy consumption [8]. It is imperative that alternative solutions to lead removal and recovery are investigated. The importance of lead removal and recovery is evident in the current lead reserves in the world. The projected term of availability of metals such as lead, copper and zinc has been determined to be 20–30 years [10].

Consortia have been demonstrated previously to have a synergistic relationship within bioremediation. In a study involving the degradation of soil pollutants, it was illustrated that more effective clean-up of pollutants was achieved with a microbial consortium compared to the single strains [11]. In a study focused on H${}_{2}$ production from poplar biomass hydrolysate, the use of a consortium was more beneficial than individual strains as the consortium had a 6.2-fold higher H${}_{2}$ production when compared to the individual strains [12,13]. In contrast, individual strains have been isolated from consortia and used for bioremediation applications. *K. pneumoniae* has previously been isolated and tested to determine its abilities to counter nitrogen pollution in wastewater. The strain was able to utilise ammonia, nitrate, and nitrite as a sole nitrogen source [14]. *P. bifermentans* has been shown to have heavy metal bioremediation potential; a *P. bifermentans* strain isolated from an estuary was able to remove 81% (0.6 μg g${}^{-1}$) of dissolved mercury(II) from solution [15]. The use of bioremediation has shown promising results, and previous work completed using a consortium obtained from a battery recycling plant in Gauteng, South Africa, has shown removals of *circa* 49% and 93% of 80 mg L${}^{-1}$ initial Pb(II) concentration after 3 h and 12 d, respectively. The same consortium removed 41% and 72% of 500 mg L${}^{-1}$ initial Pb(II) concentration after 3 h and 12 d, respectively. The dominant Pb-precipitating species was identified as *Paraclostridium bifermentans* and *Klebsiella pneumoniae* [16]. The initial removal of Pb(II) was assumed to be due to rapid adsorption or another detoxification mechanism. It was further demonstrated that the resulting precipitate included elemental Pb, which is evidence for an anaerobic respiration mechanism with Pb(II) as the terminal electron acceptor [16]. A further investigation into the biosorption of Pb(II) by the same consortium tested metabolic inhibited biomass for the removal of Pb(II) from solution and observed a 50% decline in Pb(II) concentration within 25 min. These results confirmed the presence and impact of biosorption of Pb(II) by the consortial biomass [17]. These results provided evidence for a mixed Pb(II) removal mechanism, with rapid biosorption to metabolically inactive biomass followed by a slower bioprecipitation by metabolic processes. This mixed mechanism would be most significant in industrial systems operated for extended periods in which a significant amount of non-viable biomass is accumulated, such as continuously operated systems.

The Pb(II) removal by the consortium was tested continuously. The system showed Pb(II) removal efficiencies of between 90 and 100% for inlet Pb(II) concentrations varying between 80 and 2000 mg L${}^{-1}$ Pb(II). The maximum continuous removal rate was found to be 1948.4 mg L${}^{-1}$d${}^{-1}$ [18]. These results indicate the significant potential of the system for the continuous removal of Pb(II) from the hydrosphere; however, a lack of understanding of the underlying removal mechanisms still limits the scaling of the system. To elucidate the biocatalytic contribution of the active Pb(II) precipitating species, *Klebsiella pneumoniae* was isolated from the microbial consortium [19]. Initial studies on the microbial precipitation of this strain indicated Pb(II) concentrations reached undetectable levels by the analysis equipment after approximately 63 h, with a significant decrease in Pb(II) observed in under 20 h [19].

The aims of the current study included the isolation of *Paraclostridium bifermentans* from the consortium and a comparison of the effectiveness of microbial precipitation of the two isolated strains. This research is intended to improve the understanding of the respective contributions of the active bioprecipitating strains to the overall Pb(II) removal by the consortium and consequently facilitate the future scaling of the process for an industrial application.

## 2. Results and Discussions

### 2.1. Isolation of Bacterial Strain P. bifermentans

The spread plate of the microbial consortium is shown in Appendix A, and the final streak plate containing a single strain is shown in Appendix A. This plate was analysed using 16S rDNA sequencing by Inqaba Biotech (Pretoria, South Africa), and the strain was identified as *Paraclostridium bifermentans*. The phylogenetic tree indicating the identity of the strain is shown in Figure 1.

### 2.2. Precipitation Study—Long Duration Study

In order to establish whether the two isolated strains are capable of removing and recovering Pb(II) from solution, a long duration study was conducted over 5 d using both strains. This study consisted of sampling every 24 h, with a metabolic activity determination at each sample interval. The nitrate (NO${}_{3}^{-}$) and Pb(II) concentration of each sample was also determined at each sampling interval. Three bacterial conditions were investigated: the two isolated strains and the industrially obtained microbial consortium from which the strains were isolated. This was completed to compare the performance of the pure cultures with the consortium to determine if the strains performed differently in the consortium compared to independently.

#### 2.2.1. Metabolic Activity

The metabolic activity measurements of the three experiments are shown in Figure 2. These depict a different pattern in metabolic activity for the two isolated cultures when compared to the bacterial consortium, possibly indicating that the cultures perform differently when uninhibited by other microbial strains and each other, as they would in the consortium. The two isolated strains do not differ greatly in their metabolic activity.

A peak in metabolic activity is observed for both strains at approximately 60 h, with a steady decline observed after that point. This decline could be an indication of a substrate depletion in the reactor. The point of decline for the consortium is observed much sooner than the individual strains, indicating that the consortium utilises the nutrients at a faster rate—likely due to the presence of competing species present in the system. This depletion coincides with results published on the microbial consortium from which the strains were isolated where a plateau in microbial activity is reached after 2 d [16].

#### 2.2.2. Residual Pb(II) Concentration

The Pb(II) concentration over time for each strain is shown in Figure 3. The amount of Pb(II) removed from solution by *K. pneumoniae* is slightly higher than the amount removed by the consortium, which could be an indication that *K. pneumoniae* is inhibited by other microbes in the consortium. It is observed that *P. bifermentans* is the least efficient at Pb(II) removal; however, the final removal is still significant, with a final Pb(II) concentration of approximately 13 mg L${}^{-1}$. The consortium showed a final concentration between the purified strains of approximately 4.5 mg L${}^{-1}$. What is of particular interest is the significant decrease in Pb(II) concentration immediately after inoculation in the *P. bifermentans* runs, which is a strong indication of rapid biosorption or some other detoxification mechanism onto the biomass of the *P. bifermentans*.

The rate of Pb(II) removal for *K. pneumoniae* is faster than for *P. bifermentans* and for the consortium. The plateau of the removal curve is reached at the same point as the first peak in the metabolic activity for the individual strains. The plateau of the removal curve for the consortium is reached at the same point in time as the peak in metabolic activity for the consortium. These results correlate with previous research on the continuous bioremoval of Pb(II) using the same microbial consortium from which the individual strains in this study were isolated, where a decrease in Pb(II) concentration is related to an increase in metabolic activity [18].

Both strains have the same initial concentrations of Pb(II) for the experiments, but a decrease in Pb(II) concentration is observed in the first reading. This reading was taken approximately 10 min after inoculation. The significance of this reading is shown in an enlarged portion of the first 30 min of the experiment, as shown in Figure 3. The initial drop in Pb(II) concentration could be indicative of a rapid detoxification mechanism (likely biosorption) since a drop in Pb(II) concentration is observed followed by an increase in Pb(II) concentration. The same occurrence was noted in previous research on the microbial consortium [16,20]. This is supported by previous research in which a drop of approximately 50% of Pb(II) concentration is observed after approximately 15 min using metabolically inactive biomass from the same consortium, therefore confirming the influence of biosorption [17]. The observation of a drop in Pb(II) concentration in this study can therefore be attributed to the biosorption of Pb(II) by bacterial biomass. The experiments in this study were inoculated using frozen precultures, and therefore, the likelihood of the presence of metabolically inactive bacteria is not significant. In addition, the nitrate concentrations measured at the start showed statistically insignificant differences using a t-statistic inference with a 95% confidence interval (see supplementary material for calculated t-statistic values), and this supports the assertion that the same initial Pb(II) concentrations were used. The main source of nitrates in the reactors was Pb(NO${}_{3}$)${}_{2}$, and this therefore acts as an internal standard.

The final percentage of Pb(II) removed is shown in Appendix A, which compares the effectiveness of each strain. From these data, the effectiveness of *K. pneumoniae* is highlighted once more as the concentration of Pb(II) reached levels undetectable by the analytical instrumentation after 93 h. A percentage removal of 94.4% is recorded for the consortium after 100 h, while a percentage removal of 83.8% is achieved for *P. bifermentans* after 95 h.

The visual changes observed for the two strains after the first sampling interval are shown in Appendix A. A dark precipitate is clearly observed in these images, indicating the effectiveness of microbial precipitation by both strains. A slight variance is observed in the intensity of the colour of the precipitate between the isolated strains, with *P. bifermentans* being darker. This could indicate a variance in the species of Pb(II) precipitate. The colour difference could also be an indication of the location of the Pb(II) precipitate, that is, if it is found extracellularly or intracellularly.

#### 2.2.3. Nitrate Concentration

The nitrate concentration was determined for each sample in order to establish if metabolic activity and Pb(II) precipitation are dependent on the nitrate concentration of the reactor. The nitrate concentration of the samples over 100 h can be seen in Figure 4. Initially, a decrease in nitrate concentration was observed, which corresponds to the initial increase in metabolic activity. Thereafter, the nitrate concentration remains stable for the individual strains even though a second exponential growth is observed for the individual strains. This is an indication that the metabolic activity of the strains is not dependent on the nitrate concentration. The nitrates are also not depleted for the individual strains, and it is therefore likely that nitrates are not a limiting substrate for growth in the individual strains.

The initial drop in nitrates of the individual strains is comparable to data previously published based on the microbial consortium from which the strains used in this study were isolated, where a drop of approximately 100 mg L${}^{-1}$ occurs after 1.5 d [16]. This drop is equivalent to the magnitude of the drop observed in Figure 4.

This trend differs for the consortium; an initial decrease in nitrates was observed with a plateau reached in the concentration curve after 20 h, which correlates with an increase in growth as observed between 0 and 40 h for the consortium. After this time, a drop in growth was observed, which does not correlate with the nitrate concentration, as this does not change.

### 2.3. Precipitation Study—Short-Term Study

It was concluded from the long-term study that the precipitation of Pb(II) was initiated in under 24 h, with the major metabolic activity and Pb(II) changes occurring in this time. A short-term study was then conducted with shorter sampling intervals to further analyse the precipitation mechanism of each strain. The short-term study was performed over 30 h with nine sampling intervals.

#### 2.3.1. Visual Changes

The visual changes noted during the experiment for *P. bifermentans* are shown in Figure 5. It is clear that the dark precipitate does not form in under 12 h. The visual changes noted during the experiment for *K. pneumoniae* are also shown in Figure 5, and it is clear that the dark precipitate is formed faster in these experiments than in the experiments using *P. bifermentans*. This is an indication that *K. pneumoniae* is possibly more efficient at the bioprecipitation of lead and correlates with the results observed in the long duration study. A dark precipitate did not form in experiments containing 900 mg L${}^{-1}$ initial Pb(II) concentration for both *P. bifermentans* and *K. pneumoniae*, possibly indicating that the concentration is too high for growth to occur. The experiments with 900 mg L${}^{-1}$ initial Pb(II) concentration are not discussed further.

#### 2.3.2. Metabolic Activity for Varying Initial Concentrations of Pb(II)

The metabolic activity for both *P. bifermentans* and *K. pneumoniae* for varying initial concentrations of Pb(II) is shown in Figure 6.

From Figure 6, a lag phase is observed for all bacterial strains with a longer lag phase noted for *P. bifermentans*. This could be an indication that *K. pneumoniae* is better adapted to the initial toxicity of Pb(II).

It is clear that *K. pneumoniae* reaches a higher value of peak metabolic activity than *P. bifermentans*, which could indicate that the strain is better adapted to the toxic Pb(II) environment. The strain has a faster exponential growth phase at the three observed concentrations than *P. bifermentans*, with the rate of initial growth decreasing with an increase in initial Pb(II) concentration.

*P. bifermentans* has a constant rate of growth in the initial exponential phase that is not dependent on the initial Pb(II) concentration. This could be an indication that the mechanism of precipitation remains unchanged independent of the concentration of Pb(II). An initial exponential growth is observed in *P. bifermentans*, with a second increase observed after 20 h. This correlates with previous research on the consortium and simulated LB broth, where it was concluded that the second growth is due to the increased nutrient concentrations [16].

#### 2.3.3. Residual Pb(II) Concentration

The residual Pb(II) concentration for both *P. bifermentans* and *K. pneumoniae* over 30 h is shown in Figure 7. The results indicate that *K. pneumoniae* is slightly more efficient at the bioprecipitation of Pb(II) than *P. bifermentans*. The amount of Pb(II) removed by *P. bifermentans* decreases as the initial Pb(II) concentration increases.

An initial drop in Pb(II) concentration is observed, which can be attributed to the adsorption mechanism as described in Section 2.2.2. The nitrate concentrations measured at the start of the short-term experiment also showed insignificant differences as noted by the use of the t-statistic inference. This supports the assertion that the same initial Pb(II) concentrations were used, as the main source of nitrates in the system was Pb(NO${}_{3}$)${}_{2}$. A magnified visualisation of Figure 7, which consists of the first 30 min of the experiments, depicts this drop in Pb(II) concentration and is shown in the inset figures of Figure 7.

In order to compare which strain was more efficient at Pb(II) removal, the final percentage removals of Pb(II) were compared and can be seen in Appendix A. From these data, it is clear that *K. pneumoniae* is more efficient at Pb(II) microbial precipitation than *P. bifermentans* at two of the three initial concentrations observed.

#### 2.3.4. Nitrate Concentration

The nitrate concentrations over time for *P. bifermentans* and *K. pneumoniae* are shown in Figure 8. The nitrate concentration does not approach zero, which is an indication that nitrates are not a limiting nutrient for the microbes, since the nitrates are not depleted. The microbial growth is therefore more likely dependent on the nutrient rich medium used. The initial decrease in nitrates could be attributed to an anaerobic respiration mechanism utilised by the bacteria.

In a previous study on the effects of nitrate concentration on microbial precipitation of Pb(II) using the microbial consortium from which the strains used in this study were isolated, it was concluded that the lack of nitrates in the sample has no significant effect on the bioprecipitation rate of Pb(II) in the sample [20]. It is argued that the denitrification bacteria present in the consortium and therefore responsible for the drop in nitrate concentration observed for the microbial consortium in Figure 8 are not responsible for the bioprecipitation of lead [20].

#### 2.3.5. Comparison between Metabolic Activity Measurements, Nitrate Concentration and Residual Pb(II) Concentration

The relationship between the metabolic activity, nitrate concentration and residual Pb(II) concentration is shown in Appendix A for *P. bifermentans* and in Appendix A for *K. pneumoniae*. The nitrate concentration decreases as the metabolic activity increases for all initial Pb(II) concentrations, which could indicate a relationship between metabolic activity and nitrate concentration. This decrease in nitrates is less significant for the samples containing 500 mg L${}^{-1}$ initial Pb(II) concentration for both *P. bifermentans* and *K. pneumoniae*.

There is a decrease in Pb(II) concentration observed when an increase in metabolic activity occurs for all initial Pb(II) concentrations. This could indicate that the precipitation mechanism is potentially dependent on the metabolic activity.

Both of these points are emphasised between the comparison of nitrate concentration and Pb(II) concentration, as a decrease occurs in both at approximately the same time. Further investigations are required before definite relationships can be concluded.

In order to determine if a relationship exists between Pb(II) concentration, metabolic activity and nitrate concentration, a four-part sigmoidal curve was fitted on the data to obtain the first derivative of the data using GraphPad Prism version 9.4.0. The first derivatives of the metabolic activity, nitrate concentration and lead concentration are compared to determine if a relationship exists between the entities. The fitted curves are shown in Appendix A. These curves were also used in the determination of the specific growth rate in Section 2.3.7.

From the data obtained from the fitted curves, the first derivative of each curve is calculated and plotted in Appendix A for *P. bifermentans* and *K. pneumoniae* respectively. From these curves, it is observed that there exists no clear relationship between the nitrate concentration, lead concentration and the metabolic activity.

A comparison between the data (metabolic activity, nitrate concentration and residual Pb(II) concentration) and their rates of change is observed in Appendix A for *P. bifermentans* and *K. pneumoniae* respectively. From Appendix A it is clear that the rate of metabolic activity reaches a peak as the metabolic activity increases. This trend is comparable for all initial Pb(II) concentrations investigated for *P. bifermentans* and is an indication that the initial Pb(II) concentration does not affect the metabolic growth of the organism, as stated in Section 2.3.2. This is comparable with results recorded for *K. pneumoniae*, where a similar trend is observed in Appendix A.

From Appendix A, an initial increase in the rate of change of nitrate concentration is observed. This could indicate a rapid initial uptake of nitrates in the microbe. In the case of *K. pneumoniae*, as seen in Appendix A, the peak is observed at a high concentration of nitrates, indicating that for the rest of the experiment, the microbe did not make much use of the nitrates present because the rate decreases.

The rate of change in residual Pb(II) concentration for *P. bifermentans*, as shown in Appendix A, indicates that the mechanism of bioprecipitation for *P. bifermentans* remains similar for varying initial Pb(II) concentrations. A maximum rate of change in Pb(II) concentration is observed for each initial Pb(II) concentration. This is not observed in samples containing *K. pneumoniae*, as shown in Appendix A, where the trend of the rate of change in Pb(II) concentration differs for each initial Pb(II) concentration. This could be an indication that the mechanism of precipitation for *K. pneumoniae* differs with the initial Pb(II) concentration.

The points discussed above relate to the mechanisms of precipitation utilised by the bacteria. These observations coincide with the published work by this research team on the mechanisms of precipitation where the effect of nitrates was investigated [20]. The results indicated that the usage of nitrates by the bacteria was most prominent at lower concentrations of Pb(II) and that denitrifiers present in the consortium were most likely not responsible for Pb(II) removal. This is attributed to the fact that Pb(II) removal was not inhibited by a lack of nitrates in the sample. The use of TEM and SEM in the mentioned study indicated that at lower concentrations of Pb(II), it was shown that Pb(II) precipitated extracellularly, while at higher concentrations, Pb(II) precipitated intracellularly. The intracellular precipitation was attributed to the production of metallothionein, which is a low molecular weight metal-bonding protein and is responsible for intracellular lead sequestration as well as the transportation, storage, and detoxification of lead. The occurrence of metallothionein in higher concentrations of Pb(II) was attributed to the fact that the protein is resistant to high Pb(II) concentrations. From XRD analysis, it was concluded that four compounds were present in the samples; these included: pyromorphite, lead sulphide, elemental lead, and elemental sulphur. The existence of pyromorphite was likely due to the biotransformation of PbS to pyromorphite. Smaller amounts of pyromorphite were detected in samples containing 80 mg L${}^{-1}$ Pb(II), and this could indicate that denitrifiers were present at lower Pb(II) concentrations and cause a larger amount of sulphur to be released via the enzyme nitrate reductase.

#### 2.3.6. Determination of Extracellular and Intracellular Pb(II) Concentration

The percentage composition in terms of solution, extracellular and intracellular Pb(II) concentration can be seen in Figure 9 for both strains at two different initial Pb(II) concentrations. A clear change in solution Pb(II) concentration can be seen across the time intervals, with a definite decrease occurring. This is a consequence of the microbial precipitation of Pb(II). Initially, Pb(II) is found on the extracellular portion of the bacterial cells, with this value decreasing over time. These results are indicative of an adsorption mechanism occurring within the first 6 h of the Pb(II) removal process. This adsorption mechanism was observed as a slight initial drop in the Pb(II) concentration in Figure 7.

As observed in the residual Pb(II) measurements, there is a definite decrease in Pb(II) removal with an increase in the initial Pb(II) concentration. This adsorption is attributed to a rapid detoxification mechanism and has been observed in previous research on the microbial consortium from which the strains were isolated, as discussed in Section 2.2.2.

#### 2.3.7. Specific Growth Rate

The specific growth rate of the bacteria for each experimental condition was calculated using
(1)$$\frac{dMA}{dt}=\mu \times MA$$
where $\frac{dMA}{dt}$ is the first derivative of the metabolic activity curve and μ is the specific growth rate [16,21].

These values were calculated by applying the fitted curve in Appendix A for the metabolic activity, calculating the first derivative of the fitted curve and dividing this value by the values of metabolic activity of the fitted curve [16,21]. The generation time of the bacteria was calculated using
(2)$$t_{2}=\frac{\ln(2)}{\mu_{max}}$$
where $t_{2}$ is the generation time in d, and $\mu_{max}$ is the maximum specific growth rate in d${}^{-1}$[16]. The values of μ calculated over time for each experimental condition are shown in Appendix A.

The values of $\mu_{max}$ and $t_{2}$ for each experimental condition are shown in Table 1. From these data, it can be concluded that *K. pneumoniae* has a faster exponential growth than *P. bifermentans* for initial Pb(II) concentrations of 80 and 250 mg L${}^{-1}$ but that *P. bifermentans* has a faster growth rate at an initial Pb(II) concentration of 500 mg L${}^{-1}$.

### 2.4. Precipitate Identity

The XPS profiles for the identification of the Pb precipitate are shown in Appendix A. The peaks shown in the profiles were identified with the use of the NIST X-ray Photoelectron Spectroscopy Database [22]. The results for *P. bifermentans* show that the precipitate was mainly PbS and Pb${}^{0}$. These precipitates will contribute greatly to the dark colour observed in the samples containing *P. bifermentans*. The presence of PbS is likely due to the liberation of S${}^{2-}$ ions during the catabolism of sulphur containing amino acids, such as cysteine and methionine [23].

The precipitates identified in the samples containing *K. pneumoniae* contained large amounts of PbO and either PbCl or Pb${}_{3}$(PO${}_{4}$)${}_{2}$. The absence of PbS and Pb${}^{0}$ in the precipitate of *K. pneumoniae* contributed to the lighter, browner colour observed in these samples. These results indicate that a different bioprecipitation mechanism is utilised by the strains. A previous analysis completed on the microbial consortium indicated the presence of significant fractions of PbS in runs performed anaerobically, which is an indication that *P. bifermentans* is active in the consortium [23]. The aerobic runs on the consortium contained a precipitate containing PbO and elemental Pb, which would indicate the absence of *P. bifermentans* in these runs.

## 3. Methods and Materials

### 3.1. Materials

The batch reactors and agar plates contained Luria–Bertani (LB) broth consisting of 1 g L${}^{-1}$ NaCl (Glassworld, South Africa), 20 g L${}^{-1}$ tryptone (Oxoid, UK) and 10 g L${}^{-1}$ yeast extract (Oxoid, UK), with 5 g L${}^{-1}$ agar (Sigma Life Science, Spain) added to the agar plates solution only. The stock solution of 10,000 mg L${}^{-1}$ Pb(NO${}_{3}$)${}_{2}$ consisted of 1.6 mg Pb(NO${}_{3}$)${}_{2}$ (Glassworld, South Africa) in 100 mL distilled water. Metabolic activity was determined using 3-(4,5-dimethylthiazol-2-yl)-2,5-diphenyl tetrazolium bromide (MTT) (Sigma, Aldrich, MO, USA). The nitrate concentration was determined using a nitrate test (Supelco, Germany). Ethylenedinitrilotetraacetic acid disodium salt (EDTA) (Supelco, Germany) was used in the determination of the extracellular and intracellular Pb(II) concentration.

### 3.2. Bacterial Strain Isolation

The bacterial strains were isolated from a microbial consortium obtained at a battery recycling plant in Gauteng, South Africa. *K. pneumoniae* was previously isolated using eosin methylene blue agar [19,24]. *P. bifermentans* was isolated using Luria–Bertani (LB) broth agar, and initially, several spread plates were prepared using a dilution of 10${}^{-5}$ of the bacterial consortium [25]. Thereafter, two rounds of streak plating occurred by selecting the visually desirable colonies [26].

All agar plates were grown anaerobically and then sealed in a large sterile jar with an AnaeroGen^TM^ anaerobic environment generation sachet (Thermo scientific, Basingstoke, UK) and an anaerobic indicator (Thermo scientific, Basingstoke, UK).

The isolated strains were identified using 16S rDNA sequencing by Inqaba Biotech (Pretoria, South Africa).

Once the strains were identified, batch reactors were prepared using LB broth as described in 100 mL serum bottles with distilled water. The reactors were inoculated using a sterile loop and a single colony from the final agar plate. The precultures were then stored in sterile 2 mL vials containing 20 v/v% glycerol and kept at −40 ${}^{\circ }$C [16].

### 3.3. Batch Reactor Preparation

The occurrence of lead precipitation was studied in batch reactors prepared in triplicate with LB broth as the nutrient media. The LB broth used was modified to minimise the reaction between NaCl and Pb(NO${}_{3}$)${}_{2}$ producing insoluble PbCl${}_{2}$. The reactors were made up in 100 mL serum bottles with distilled water. The required amount of Pb(NO${}_{3}$)${}_{2}$ was added to the mixture to produce varying concentrations of Pb(II). Four initial Pb(II) concentrations were investigated in this study: 80 mg L${}^{-1}$, 250 mg L${}^{-1}$, 500 mg L${}^{-1}$ and 900 mg L${}^{-1}$.

The reactors were inoculated with the precultures prepared in Section 3.2 by adding 0.2 mL of the defrosted preculture to the reactor. The reactors were then purged with N${}_{2}$ for 3 min to achieve anaerobic conditions. Sampling was performed using a sterile hypodermic needle and syringe, and the sample was dispensed in sterile 2 mL vials and stored at −40 ${}^{\circ }$C.

### 3.4. Metabolic Activity

The metabolic activity was determined using MTT, which is a water-based yellow dye. The metabolic activity measurements were taken using a UV-vis spectrometer at 550 nm (WPA, Labotec, Midrand, South Africa) with a previously described method [16,27]. As a preliminary study, colony-forming unit (CFU) plates for the respective strains were grown and correlated with the corresponding metabolic activity measurements. The results shown in Figure S1 clearly demonstrate the correlation between CFU and metabolic activity, thereby validating the use of metabolic activity as a measure of growth.

### 3.5. Nitrate Concentration

The concentration of NO${}_{3}^{-}$ was determined with the use of a nitrate test, in which the nitrates react with a benzoic acid derivative in concentrated sulfuric acid to form a red nitro compound that is determined photometrically (Spectroquant NOVA 60, Merck, New York, NY, USA) [16]. Samples were prepared by centrifugation at 7711× *g* for 10 min, and the supernatant was analysed.

### 3.6. Lead Concentration

The concentration of Pb(II) ions in solution was determined by centrifuging the sample at 7711× *g* for 10 min. The supernatant was diluted in distilled water, and the Pb(II) concentration was determined with the use of an atomic absorption spectrometer with a Pb lumina hollow cathode lamp (PerkinElmer AAnalyst 400, Waltham, MA, USA).

### 3.7. Determination of Extracellular and Intracellular Pb(II)

The extracellular and intracellular concentration of Pb(II) was determined using a combination of previously described methods [28,29,30]. The samples were centrifuged at 7711× *g* for 10 min, and the supernatant Pb(II) concentration was determined. The pellet was then dissolved in 1 mL 20 mmol L${}^{-1}$ EDTA solution with a contact time of 30 s. The sample was then centrifuged at 7711× *g* for 10 min and the supernatant concentration, which is the extracellular Pb(II) concentration, was determined as described in Section 3.6. The pellet was then dissolved with 55% HNO${}_{3}$ solution, and the intracellular Pb(II) concentration was determined. A description of the process along with the expected concentrations and expected species obtained at each step is shown in Figure 10.

### 3.8. Precipitate Identity

The precipitate samples were characterised by X-ray photonelectron spectroscopy (Thermo ESCAlab 250, Xi, Waltha, MA, USA) [16]. Sample preparation involved the centrifugation of the 2 mL samples at 7711× *g* for 10 min. The samples were then dried anaerobically to ensure that minimum oxidation of the precipitate occurred. The drying process involved placing the samples in a sealed jar containing silica crystals and an AnaeroGen^TM^ anaerobic environment generation sachet (Thermo scientific, Basingstoke, UK) for 24 h. The XPS data were deconvoluted using OriginPro 2022 (OriginLab Corporation, Northampton, MA, USA) and analysed using the NIST CPS database [22].

## 4. Conclusions

The bioprecipitation abilities of two strains isolated from an industrially obtained microbial consortium were compared in both a short- and long-term study over 30 h and 5 d, respectively. The strains identified as *P. bifermentans* and *K. pneumoniae* removed approximately 86% and 91% of Pb(II) from solution, respectively, in experiments containing 80 mg L${}^{-1}$ initial Pb(II) concentration after 30 h. The same strains were able to remove approximately 84% and 100% of Pb(II) from solution in experiments containing 80 mg L${}^{-1}$ initial Pb(II) concentration after approximately 5 d, respectively.

It was concluded that the residual Pb(II) concentration was not dependent on the metabolic activity of the sample, and that metabolic activity was not dependent on the amount of nitrates present in the sample. It was determined that the nitrate concentration was not a limiting substrate and that the growth of the microbes is most likely dependent on the concentration of the constituents of the nutrient broth.

The extracellular and intracellular concentrations of Pb(II) were determined for the samples, and this indicated the presence of an initial detoxification mechanism such as biosorption in the first 6 h of the experiments.

The identities of the precipitates obtained for each strain vary, with PbS and Pb${}^{0}$ observed as the main precipitates present in samples containing *P. bifermentans*, and PbO with either PbCl or Pb${}_{3}$(PO${}_{4}$)${}_{2}$ present in samples containing *K. pneumoniae*.

## Figures and Tables

**Figure 1 ijms-23-12255-f001:**
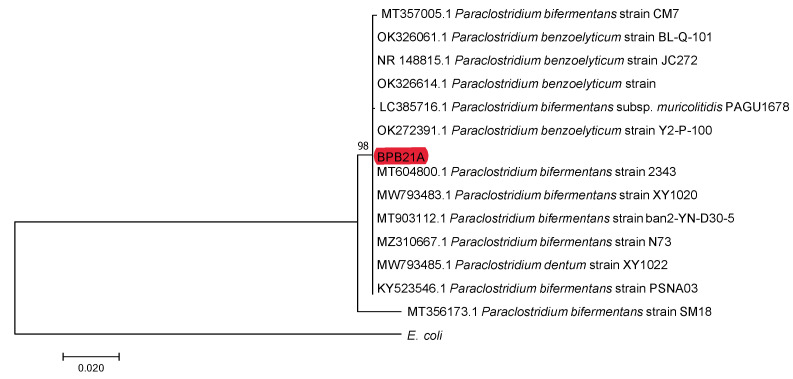
Phylogenetic tree indicating the identity of the isolated *Paraclostridium bifermentans* strain (BPB21A).

**Figure 2 ijms-23-12255-f002:**
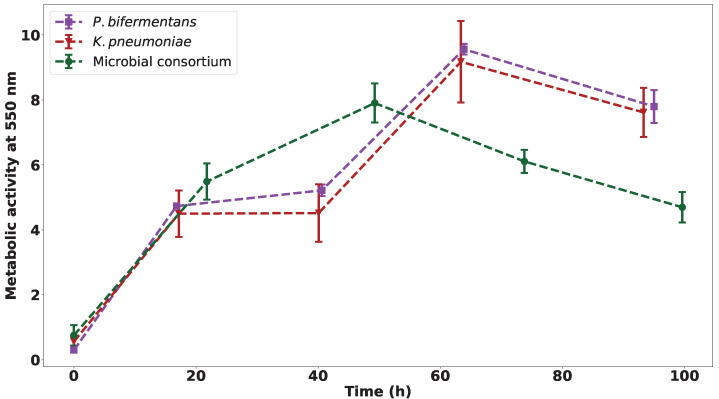
The metabolic activity of the pure cultures and the microbial consortium observed over 100 h, with *n* = 3.

**Figure 3 ijms-23-12255-f003:**
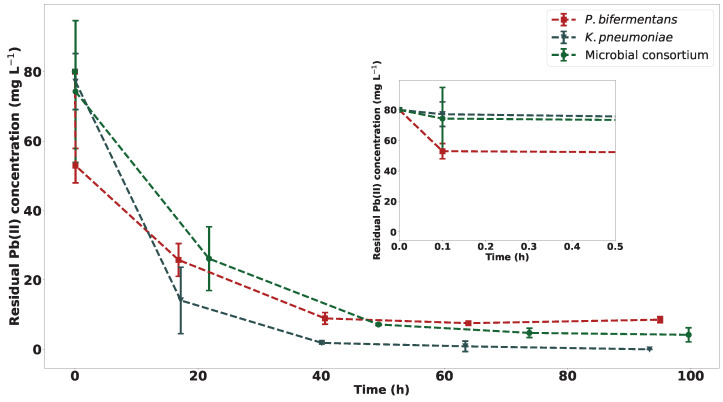
The residual Pb(II) concentration change over time for each experiment over 100 h with *n* = 3, with an inset graph depicting the initial 30 min of the experiment.

**Figure 4 ijms-23-12255-f004:**
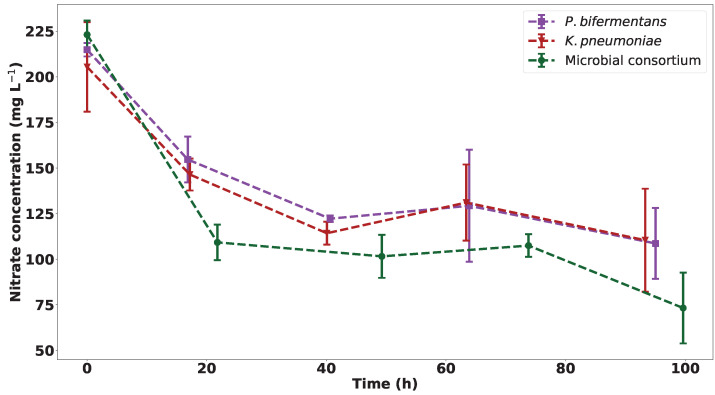
The nitrate concentration of the samples in the three experimental runs over 100 h, with *n* = 3.

**Figure 5 ijms-23-12255-f005:**
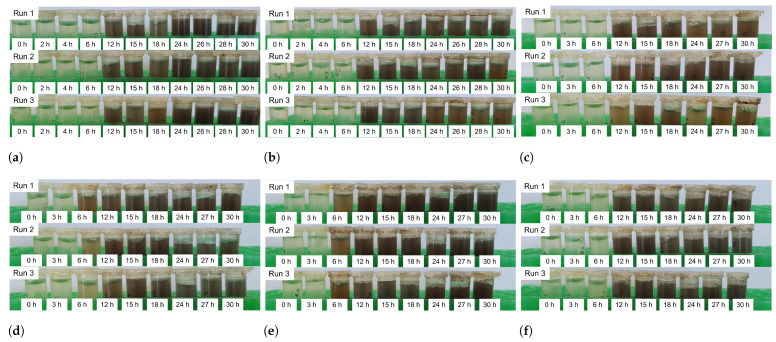
The visual results for samples containing *P. bifermentans* at (**a**) 80 mg L${}^{-1}$, (**b**) 250 mg L${}^{-1}$ and (**c**) 500 mg L${}^{-1}$ initial Pb(II) concentration and samples containing *K. pneumoniae* at (**d**) 80 mg L${}^{-1}$, (**e**) 250 mg L${}^{-1}$ and (**f**) 500 mg L${}^{-1}$ initial Pb(II) concentration.

**Figure 6 ijms-23-12255-f006:**
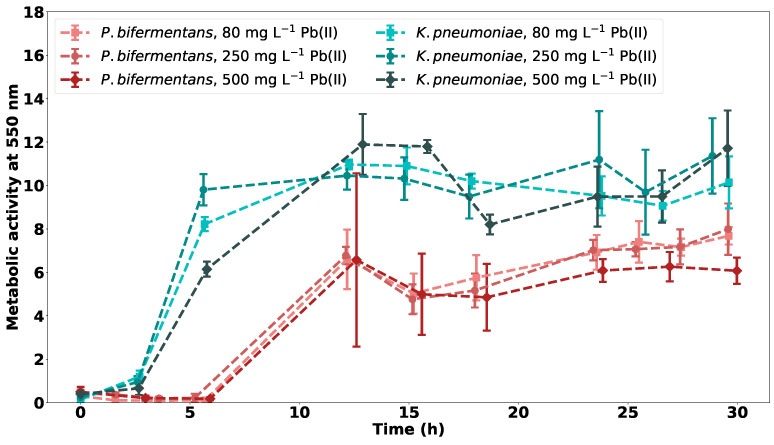
The metabolic activity at 550 nm for *P. bifermentans* and *K. pneumoniae* at varying initial Pb(II) concentrations, with *n* = 3.

**Figure 7 ijms-23-12255-f007:**
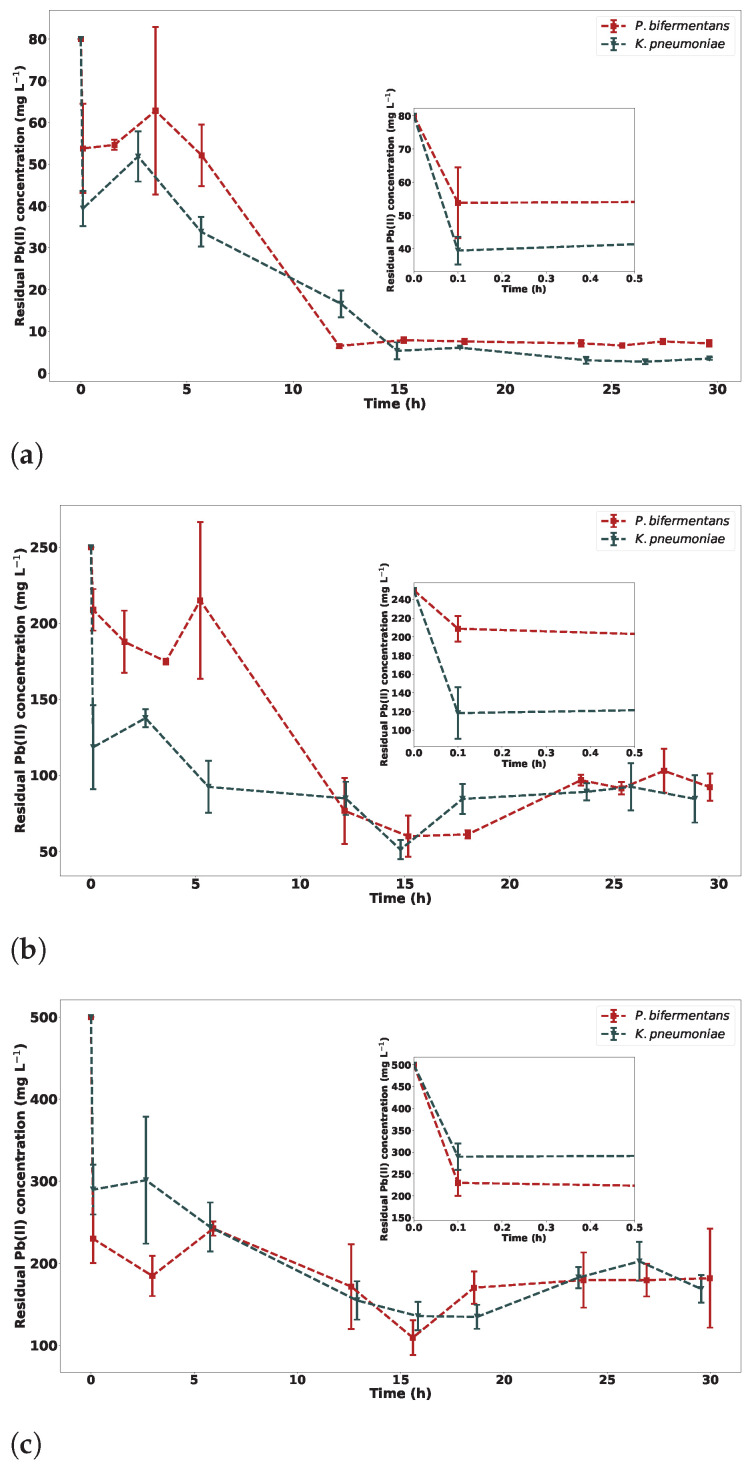
The residual Pb(II) concentration for *P. bifermentans* and *K. pneumoniae* at (**a**) 80 mg L${}^{-1}$, (**b**) 250 mg L${}^{-1}$ and (**c**) 500 mg L${}^{-1}$ initial Pb(II) concentration with *n* = 3, with an inset graph depicting the initial 30 min of the experiment.

**Figure 8 ijms-23-12255-f008:**
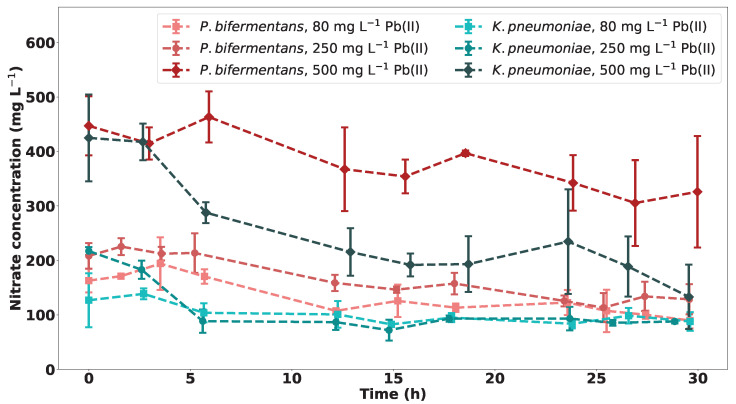
The nitrate concentrations over time for experiments of varying initial Pb(II) concentrations for both *P. bifermentans* and *K. pneumoniae* with *n* = 3.

**Figure 9 ijms-23-12255-f009:**
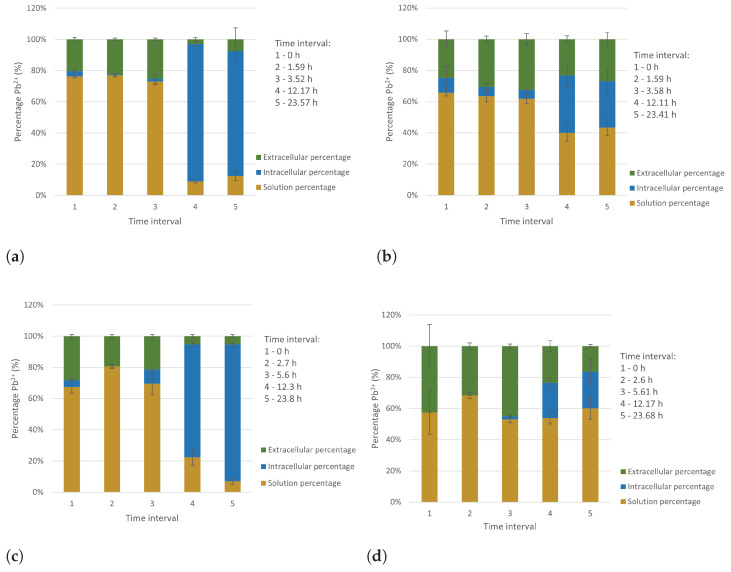
The percentage composition of the samples in terms of extracellular, intracellular and solution Pb(II) for (**a**) 80 mg L${}^{-1}$ initial concentration Pb(II) with *P. bifermentans*, (**b**) 250 mg L${}^{-1}$ initial concentration Pb(II) with *P. bifermentans*, (**c**) 80 mg L${}^{-1}$ initial concentration Pb(II) with *K. pneumoniae* and (**d**) 250 mg L${}^{-1}$ initial concentration Pb(II) with *K. pneumoniae*.

**Figure 10 ijms-23-12255-f010:**
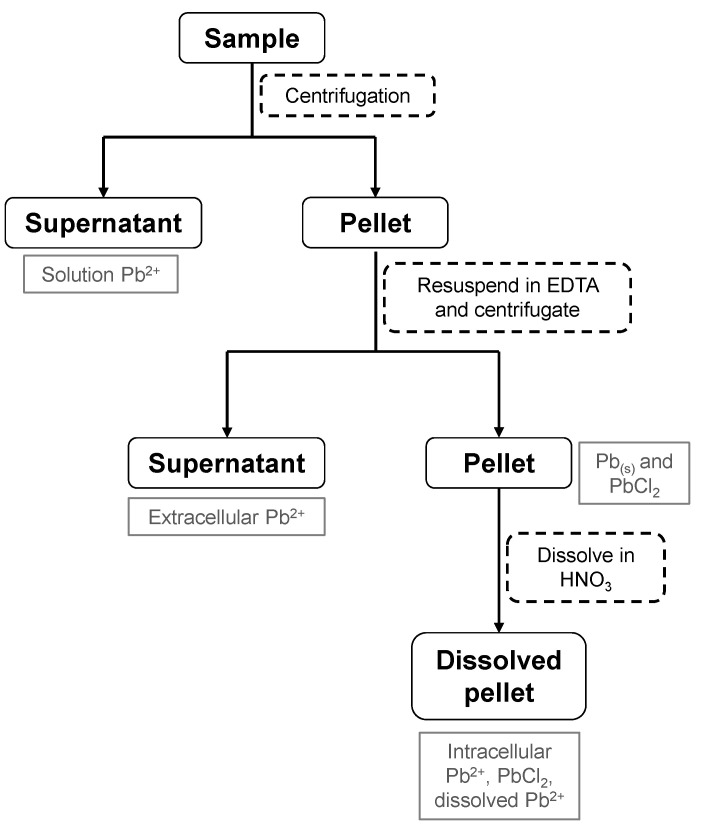
A description of the process to determine extracellular and intracellular Pb(II) concentration as well as the other possible lead species present in the steps.

**Table 1 ijms-23-12255-t001:** The value of the maximum specific growth rate ($\mu_{max}$) and the generation time ($t_{2}$) for all the experimental conditions observed for both strains.

Strain	Initial Pb(II) Concentration (mg L${}^{-1}$)	$\mu_{\max}$ (d${}^{-1}$)	$t_{2}$ (d)
*P. bifermentans*	80	7.94	0.0873
	250	15.2	0.0441
	500	280	0.00247
*K. pneumoniae*	80	36.1	0.0192
	250	45.6	0.0152
	500	87.2	0.00795

## Data Availability

The data presented in this study are openly available in the University of Pretoria Research Data Repository at doi: 10.25403/UPresearchdata.21324492.

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
