# Peer review of "Microbial Precipitation of Pb(II) with Wild Strains of Paraclostridium bifermentans and Klebsiella pneumoniae Isolated from an Industrially Obtained Microbial Consortium"

_ijms, 2022, doi:10.3390/ijms232012255_

Round 1

Reviewer 1 Report

The authors present a study aiming to compare the precipitation of Pb by P. bifermentans and K. pneumoniae isolated from an microbial consortium. Nevertheless, figure 5, 6, 9 and 10 compare data from present study with the data from published previously (reference 9 and 12). Can different strains tested at different experimental conditions (different experimental days, different laboratory researcher, etc…) be compared? Moreover, the authors don’t refer the n for any result, except for figure 7 and 8. Data from table 2 is derived from figure 10 but the initial Pb values are not the same. Additionally the removal percentages are calculated in this table 2 and used in conclusions and these values don't gave confidence to accept this study for publication.

Reviewer 2 Report

The manuscript by Sekoai et al. “Microbial Precipitation of Pb(II) with Wild Strains of Paraclostridium bifermentans and Klebsiella pneumoniae Isolated From an Industrially Obtained Microbial Consortium,” demonstrated the microbial precipitation potential of bacterial derived from Pb(II)-resistant microbial consortium of industrial origin. The manuscript is noteworthy and requires substantial revision to justify its publication.

Comments

1.      The authors should delete the use of standard deviation (SD) in the abstract or in the main text.

2.      The mechanism of microbial precipitations involving few biocatalytic activities i.e., reductases, should be illustrated and discussed with additional data in the manuscript.

3.      Introduction, too many short paragraphs; please combine them into two-three paragraphs, only.

4.      Lines 20-31, please provide the quantum of lead in wastewater.

5.      Lines 41-46, please briefly highlight the importance of consortia for biological applications, including bioremediation. i.e., the positive interactions (synchronized behavior) among microbial populations is a crucial approach to developing success consortia or establishment. In addition, please highlight or discuss the role of Klebsiella (a newly recognized name of Enterobacter) in the interactions with other microbes to justify its significance. i.e., International Journal of Hydrogen Energy 39 (2014) 14663-14668; Journal of Cleaner Production 287 (2021) 125037.

6.      Discussion is week. Please properly discuss the finding with recent citations to justify the significance (minor, in the sections such as 3.2, 3.2.2-3.2.4, 3.3.2-3.3.6, and 3.4).

7.      Too many Figures in the main text. Please provide informative data as 6-7 figures only in the main text; others should be added as supporting files. Also, improve the Figure's quality i.e., font size, line width, and resolution, etc.

Round 2

Reviewer 1 Report

The authors explained the issue pointed in previous revision. Nevertheless, the authors present a study aiming to compare the precipitation of P. bifermentans and K. pneumoniae isolated from a microbial consortium. But, they compare data from present study with the data from published previously. Again, can different strains tested at different experimental conditions (different experimental days, different laboratory researcher, etc…) be compared? Additionally the removal percentages reported in conclusions are calculated based on these types of assays. I am not confident to accept this paper for publication.

Major Revisions

Figure 4, 5, 8 and 9 compare data from present study with the data from published previously (reference 9 and 12). However, I am not I confident that compare different strains at different experimental conditions is the correct form to do it. Why weren’t the strains tested at the same experimental conditions (same experimental days, same laboratory researcher, etc…). The authors should answer this main point.

In figure 5 the residual concentration at time zero is not the same for K. pneumonia and for P. bifermentans, please explain why. Also, clarify what is the initial residual Pb concentration in this figure. Explain how the previous point interferes with the kinetic for each strain. Please give example of other studies published by other authors in scientifically reliable journals that use a similar strategy.

It is possible to compare different strains when they are tested at different days? This means, the authors have published results for K. pneumoniae (reference 12 and 9) and they now want to compare those published results with the results obtained by P. bifermentans. Again, I am not confident that this is the correct form to do it. Please give example of other studies published by other authors in scientifically reliable journals that use a similar strategy.

Minor Revisions

English must be revised

Methods used in previous works should not be mentioned in introduction

Methods

A section of materials/chemicals/biological should be introduced as first section in methods that will be the adequate place to put the references of products. This will avoid confusing text when describing the methods

Results

Section 3.2 is not a result but a method, also section 3.2.1 doesn’t show a main result and doesn’t justify to be considered a section.

In figure 5 the data from K. pneumonia was not used from previous studies? Explain why.

In figure 5 at 24h and 30h the precipitation of Pb(II) is still occurring, nevertheless the authors used these time points for short precipitation studies. Please clarify when is scientifically considered that precipitation occurs so that the following sentence can be considered correct “It was concluded from the long term study that the precipitation of Pb(II) occurs in under 24 h.”

Join figure 6 and 7 in only 1 figure

General for all figures - Improve quality to facilitate visualization by increasing letter and using bold definitions. Same for graphical points.

Reviewer 2 Report

Accept as is

Author Response

The authors would like to thank the reviewer for the time spent reviewing the manuscript. 

Round 3

Reviewer 1 Report

The authors greatly improved the manuscript visualization and answered to the reviewers comments.Nevertheless, figure 10 was changed but there was a lack of attention to maintain it appealing for the reader. This figure should be improved.

Author Response

The authors would like to thank the reviewer for the valuable input received during the review process. Regarding Figure 10, the zoomed portion of the residual Pb(II) concentration graph was added to the residual Pb(II) graph as an inset figure, with the intention of making the graphs easier to understand. This was done for Figure 4 as well.